# Generation of severely scoliotic subject-specific musculoskeletal models

**Samuele Luca Gould**[1⊕], **Giorgio Davico**[1⊕], **Monica Cosentino**[2⊕], **Luca Cristofolini**[1⊕*], **Marco Viceconti**[1⊕]

**1** Department of Industrial Engineering, Alma Mater Studiorum - University of Bologna, Italy, **2** Medical Technology Lab, IRCCS Istituto Ortopedico Rizzoli, Bologna, Italy

⊕ These authors contributed equally to this work.
* luca.cristofolini@unibo.it

## Abstract

Surgical correction of severe scoliosis has been associated with high rates of complications and reoperation rates. Patient outcomes could be improved with the use of personalised scoliotic spine models. To be clinically applicable, these models must be robust to operator variability, must accurately reflect the patient anatomy, and be rapidly generated. This study developed a semi-automatic pipeline for creating subject-specific models, and the inter- and intra-operator variability and model accuracy were assessed. An existing generic spine model was modified and morphed into a subject-specific spine model using manually performed virtual anatomical landmark palpations through a semi-automatic pipeline. The inter- and intra-operator variability of the virtual palpations was assessed, and the model was compared to the ground truth of a radiographic evaluation and segmentation of computed tomography data. The interclass correlation coefficient showed excellent inter- and intra-operator repeatability (>0.9). Although significant differences were found, they were not associated with specific anatomical landmarks. The mean inter- and intra-operator variability of the virtually palpated anatomical landmarks was 2 mm, and the maximum was 10.3 mm. The vertebral centres of the patient-specific model had maximum median errors of 10.3 mm. While the mean curvature of the model reflected the radiographic measurements, there were substantial deviations from the mean. The semi-automatic pipeline successfully created a subject-specific scoliotic spine model that included automatically adjusted muscle paths. The results indicated the process was robust to inter- and intra-operator variability but would benefit from full automation, particularly to improve the definition of the intervertebral joint.

## 1. Introduction

Severe scoliosis may require surgical treatment to restore pulmonary and cardiac function, improve quality of life, and reduce back pain and the risk of future disability

**Data availability statement:** All data and code files are available from the figshare database (https://figshare.com/s/9bc5c72fb94cf3487ab2).

**Funding:** Funding from the European Health and Digital Executive Agency, HEU H2022 project "Metastra - Computer- Aided Effective Fracture Risk Stratification Of Patients With Vertebral Metastases For Personalised Treatment Through Robust Computational Models Validated In Clinical Settings" (topic HLTH-2022-12-01, grant ID 101080135). URL: https://hadea.ec.europa.eu/index_en The funders had no role in study design, data collection and analysis, decision to publish, or preparation of the manuscript.

**Competing interests:** The authors have declared that no competing interests exist.

[1,2]. These surgeries are complex, and aside from sagittal spine alignment being the most critical priority, the aims of the surgery may vary greatly between surgeons, which also results in a large variation of instrumentation strategies [3]. The treatment strategy is not necessarily selected based on the predicted post-operation function [4]. Reported scoliotic surgery complication rates vary widely, from 0% to 89%, depending on the type of scoliosis, time post-operation, and the definition of complication [2]. Reports include a follow-up rate of 9.9% for major complications and overall 6% requiring reoperation within ten years of the original operation [5]. Although other studies have found reoperation rates as high as 46.7% within two years [6]. Many factors can increase the risk that a reoperation is required, including time post-operation, age of operation, and surgical treatment technique [6–10]. Computational models may help guide surgeons to objectively select the optimal procedure for complex spine correction surgeries by simulating virtual instrumentation strategies and predicting treatment outcomes [4,11], which would potentially improve surgical outcomes and reduce complication rates. However, this requires suitable computational models.

Musculoskeletal multibody (MSK) modelling techniques have been used to simulate the scoliotic spine [12,13] and scoliotic surgical corrections [14,15], for example, identifying optimal instrumentation techniques and complication risks [13,16]. This has been done with commercial software such as AnyBody [13] and open-source software, such as OpenSim [12]. Open-source software allows for models to be shared and reused, making them more accessible [17,18]. Some generic open-source models of the healthy spine already exist [19,20]. However, for such models to be clinically useful, they require personalisation [4,17].

The study by Bruno et al. created generic male and female models with a fully articulated spine [19,21]. Each intervertebral joint (IVJ) from the thoracic segment (T1) to the sacrum included three rotational degrees of freedom (DoF): lateral bending, axial rotation, and flexion-extension. Additionally, the ribs (with costovertebral joints) and sternum, and the major trunk muscle groups were included [19]. From these models, models of adolescents from 6 to 18 years old were created using a non-linear scaling technique [20]. One recent study has developed an automated pipeline using deep learning to create a musculoskeletal model of the lumbar spine [22].

Other studies have developed pipelines to create mildly and severely scoliotic models in AnyBody [13,23] and mildly scoliotic models in OpenSim [12]. However, they have often relied on manual segmentation of computed tomography (CT) data, or geometric reconstruction and data from biplanar images [15, 23, 24](EOS Imaging Systems, Paris, France [25]). Overbergh et al. also developed a pipeline to create subject-specific scoliotic models, which too relied on manually segmented subject-specific vertebral geometries from CT data [24], similar to other studies which used data from bi-planar X-rays [14]. Manual segmentation is a time-consuming process [22]. The 2019 review by Galbusera et al. identified the application of artificial intelligence (AI) and machine learning for the development of high-quality automatic segmentation methods, noting that the performance of existing AI approaches could

still improve substantially [26]. The number of recent studies on the topic suggest that investigation into high quality automatic segmentation tools is ongoing [27–32]. Two common approaches used in automatic segmentation are statistical shape modelling [32] and deep learning [27–31]. However, both suffer with the drawback of requiring large clinically representative datasets to develop the automatic segmentation tools; the availability of such datasets is limited (especially of severely scoliotic cases) [27,33]. Additionally, the training of these models is often associated with high computational memory requirements and computational cost [28].

The need for large clinically relevant datasets currently limits the applicability of such approaches in more complex cases such as vertebral fractures, scoliosis, and osteophytes [27,29,31,33]. Complex geometries, such as vertebrae with osteophytes, have led to errors in segmentations [29], in particular for severe scoliosis the automatic segmentation has been suboptimal [30] compared to cases where the spinal alignment in the frontal plane is maintained [31]. Furthermore, to the best to the author's knowledge, such approaches for cases of severe scoliosis have not yet been incorporated into the larger pipelines that are required to create MSK models which would also require identification of attachment points, joint locations, and muscle paths.

Alternatively, semi-automatic methods for creating scoliotic spine models recreate the subject-specific curvature using virtual palpation to identify anatomical landmarks [23], or marker clusters placed on the subject [24]. However, these do not account for bony geometries. Further, scoliotic models including the musculature have been used to investigate muscle activation, shear loading, and compressive forces conditions within the IVJ [12,13]. Other models have simulated surgical corrections, but not included musculature [15]. However, to the best of the author's knowledge, a pipeline compatible with OpenSim to create severely scoliotic subject-specific spine models from CT data and virtual anatomical landmark palpations is not available.

Similar to studies which have shown the importance of lower limb model sensitivity to inter- and intra-operator variability of virtually palpated landmarks [34,35], the sensitivity of spine models to inter- and intra-operator variability should be considered. Studies focusing on the spine have investigated the sensitivity of spine curvature and scaling of personalised healthy spine models to the between-session variability of reflective marker placement [36]. However, such studies need to be extended to include pathological cases [24]. Furthermore, this does not account for virtually palpated anatomical landmarks' inter- and intra-rater variability. This is a necessary step for pathological models to be considered reliable and representative of an individual [35,37] and is especially relevant if the models are to be used in clinical applications [17]. Furthermore, the variability of the virtual palpation of anatomical landmarks in scoliotic cases could be expected to increase compared to a non-pathological spine, as the deformity of scoliotic vertebrae would likely make identifying the landmarks more difficult.

The sensitivity of creating personalised scoliotic models to inter- and intra-rater variability has been considered when an EOS imaging system is used [38], however to the best of the author's knowledge, no study has investigated the impact of the variability of virtually palpated landmarks on creating personalised scoliotic models. This could be because the EOS imaging system, with the dedicated software, allows for the rapid (~10 minutes) creation of personalised models through three-dimensional (3D) reconstruction [39] without virtual palpations. However, while the EOS imaging system is considered the gold standard [40], with over 500 EOS imaging systems installed worldwide, they remain less widely available than CT scanners [41].

In addition to the curvature of the scoliotic spine, the definition of soft tissues such as the intervertebral disc, ligaments and muscles should be considered when creating MSK models of the spine. Including these tissues is required to estimate the necessary corrective forces and the forces and moments the instrumentation may experience post-operation. Both the rotational and translational stiffnesses of the IVJ play an essential role in the kinematics of the spine [42]. Schmid et al. incorporated non-linear stiffnesses in the rotational DoF, but not in translation [12]. Bassani et al. included stiffnesses in translational DoF; however, the version of the software used in the study did not allow for stiffnesses to be defined in the rotational DoF [23]. Stiffnesses of the IVJ may play an essential role in the effectiveness of a surgical correction for

scoliosis [43], meaning their inclusion in models for predicting surgical outcomes is essential. Other studies which have included stiffness in all DoF have not included the musculature [14,15]. Considering the muscles, muscle parameters are assumed to be the same for scoliotic cases as for non-pathological cases [13]. The muscle paths are modelled as straight lines, based on attachment and insertion points [19,23]. However, a straight path may be inaccurate in severe scoliosis, where sagittal misalignment may affect the muscle paths. The use of wrapping surfaces and via points may result in a more accurate muscle path [44], however, such techniques have not yet been incorporated into scoliotic spine models.

Therefore, this study aimed to: (i) develop an opensource package which provides the framework to generate scoliotic MSK models in OpenSim with subject-specific curvature, from CT data, while also automatically adapting muscle paths to follow the spinal curvature and allowing for the simple implementation of personalised IVJ stiffnesses; (ii) to assess the robustness of the method to inter- and intra-operator variability; (iii) to quantify the accuracy of the method as closeness to a ground truth.

## 2. Materials and methods

To provide a brief overview of the methodology:

1. This study first developed a base model. The base model was developed from an existing spine model by removing and adding various components (for example, a statistical shape model of the sacrum was added).

2. Next, this base model was morphed into a subject-specific model using CT data (this subject was not used in developing the base model). Anatomical landmarks in the CT data were virtually palpated to perform the morphing.

3. Virtual palpation introduces operator dependency; therefore, the inter- and intra-operator variability was assessed.

4. Finally, the accuracy of the subject-specific model was assessed by comparing it to data derived from the CT and X-ray data (metrics obtained from a radiographic evaluation and a segmentation of all the vertebral bodies).

### 2.1. Ethics

The patient was enrolled for an observational study approved by the local Ethics Committee of Istituto Ortopedico Rizzoli on 01/07/2019 (protocol number 0007902) and signed the study-specific informed consent to allow the collection of pseudo-anonymized clinical data and imaging from the Hospital digital archive. The study was conducted according to the principles of the Declaration of Helsinki.

The data (specifically CT data, X-ray data, and patient age and sex) was accessed between the dates 07/10/2021 and the 13/04/2022.

### 2.2. Development of a base model

The OpenSim MatLab API 4.1 [18,45] for MatLab (v R2022b) with a custom script was used to create the base model, which was developed from the generic full-body female model with a fully articulated spine [19]. First, some simplifications were made, certain bones and muscles (modelled as actuators) were removed, and the existing marker set was removed. This left the vertebrae (T1 to L5), the sacrum, and the pelvis. Muscles between the remaining bodies were kept. After this, parts of the model were enhanced. The original IVJs that allowed for the three rotational DoF were modified to also allow for the three translational DoF (for a total of 6 DoF at each IVJ). A lumped parameter spring-damper element (referred to in OpenSim as a bushing element) was then added at each IVJ, in the same relative location as the joints, to model its mechanical properties, as done by Christophy et al. (2012) [46]. It has a linear spring-damper element defined in each DoF (Table 1), and stiffness and damping parameters were assigned based on the literature values [47]. This simplification was made as the properties in scoliotic spines may be highly subject-dependent, and using any particular values would add little benefit. Notably, the proposed pipeline includes a function enabling seamless personalisation of any DoF's stiffness properties.

**Table 1. Stiffness and damping parameters used to define the IVJ.**

| Degree of freedom | Stiffness | | Damping | |
|---|---|---|---|---|
| | Translation, N/m | Rotation, Nm/rad | Translation, N/(m/s) | Rotation, Nm/(rad/s) |
| Anterior-Posterior Shear/ Right-Left Bending | 149,000 | 68.8 | 1,000 | 2.3 |
| Inferior-Superior Translation/ Axial Rotation | 1,890,000 | 291 | 1,000 | 2.3 |
| Right-Left Translation/ Flexion-Extension | 135,000 | 51.0 | 1,000 | 2.3 |

The sacrum on the original model lacked sufficient detail to allow a virtual palpation of anatomical landmarks. Therefore, CT data of 18 sacra, which included the coccyx region, were retrieved from open-access datasets on the Cancer Image Archive (https://www.cancerimagingarchive.net/) [48–52]. The CT data was from patients with lymphadenopathy or prostate or anal cancer. Other patient data were not available as the images are de-identified. The CT data was manually segmented in Mimics Innovation Suite v24 (Materialise, Leuven, BE) and exported as .stl files. These files were cleaned, aligned, and converted to a .wrl file in MeshLab [53]. A statistical shape model was created with the GIAS2 Shape Modelling software [54]. This allowed for the virtual palpation of the necessary anatomical landmarks and maintained a generic representation of the sacrum.

The script to create the base model replaced the sacrum's original geometry with the mean geometry of the statistical shape model, the other bones used the original vertebrae .vtp files.

A marker set was required to create a scoliotic model from the base model. The markers set was created by a virtual palpation of the vertebral geometries (the statistical shape model and the original .vtp files) in NMS Builder [55] and then added programmatically.

The muscle paths in the base model were direct lines between attachment points. Therefore, wrapping spheres were introduced to avoid in-bone penetration when creating the scoliotic spine model. Wrapping spheres were manually added to each vertebral body. Repurposing the code by Modenese et al. [56], the radius and position of each wrapping sphere were determined by fitting a sphere with a least squares minimisation to the anterior surface of the .stl files of the vertebral bodies (Fig 1). This was assumed to approximately correspond to the centre of the vertebral body.

### 2.3. Generation of a subject-specific scoliotic model from CT data

**2.3.1. Processing of the CT data.** The CT data were manually segmented in Mimics (Fig 2), this provided a ground truth geometry. Segmentation of a single vertebra required approximately an hour. The centres of the vertebral bodies were estimated using the same least squares minimisation methodology (Fig 1).

A protocol defined two groups of virtual landmarks (S1 File), one to define the joint poses and the other to enable scaling (Fig 3), which are palpated on the CT scan combined with a simple segmentation of the scan achieved with a single, quick thresholding operation with no further processing. Locations for the markers used to define the joint poses were selected based on the recommendations from the International Society of Biomechanics (ISB) [57]. In brief, the axis defining right-left translation direction is defined by the direction of the line between the landmarks of the pedicles of the vertebrae; the axis defining the inferior-superior translation direction is defined by the lines that connect the landmark on the centre of the inferior and superior endplate of the vertebrae; the axis defining the anterior-posterior shear is then perpendicular to the other two axes; and the origin of the joint is the intersect of the lines passing through the landmark on the centre of the inferior and superior endplate of the two adjacent vertebrae.

Locations of the markers for scaling were chosen based on locations used in previous studies [24,58]. Additionally, a naming convention was applied to be compatible with the code used to create the scoliotic model and the markers on the model.

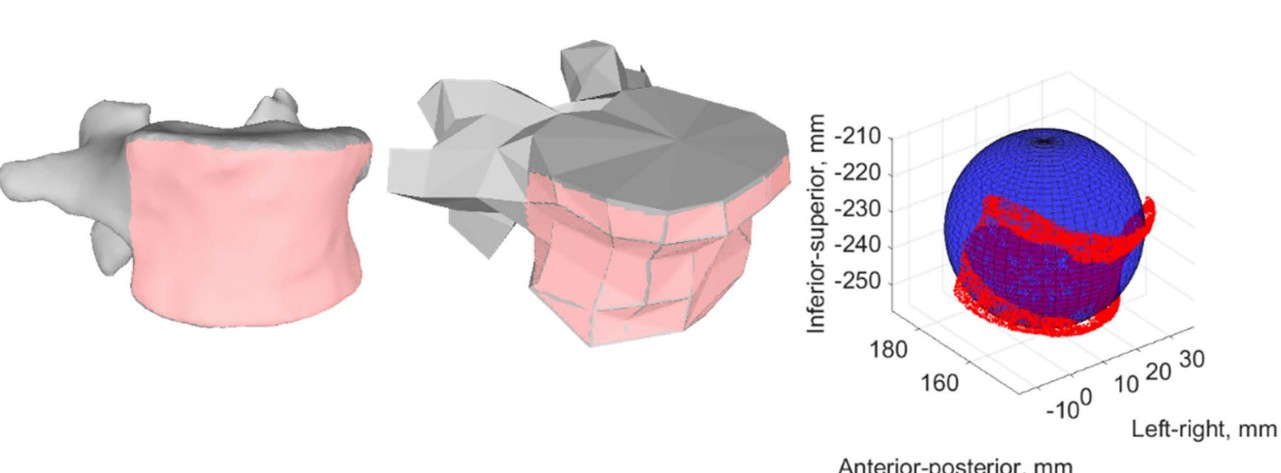

**Fig 1. Fitting of spheres to the anterior surface of the vertebrae.** The segmented (left) and generic (centre) .stls of L5 and the selected nodes on the anterior surface (red). The sphere (blue) fitted to those nodes using a least squares distance algorithm (right).

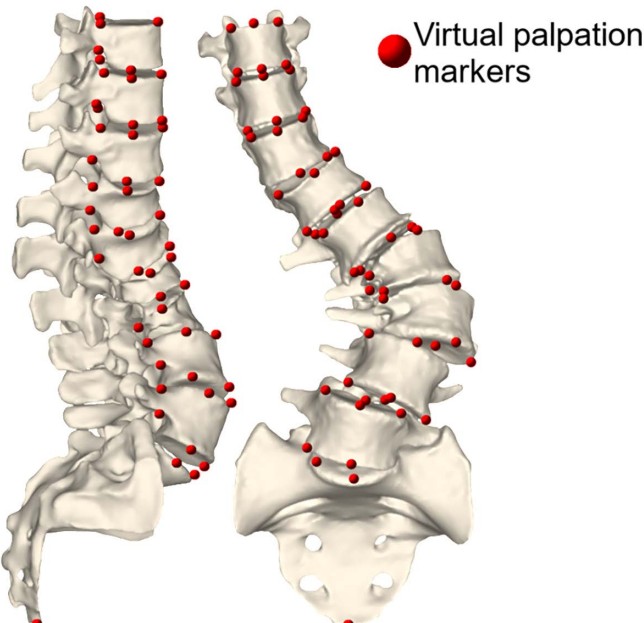

**Fig 2. Segmentation of the CT data.** Sagittal (left) and frontal (right) view of the geometries from the segmentation of the CT data, the sacrum and vertebrae L5 to T9 were segmented. The positions of the virtual palpation markers that are visible in these views are also shown by the red spheres. The palpations were performed on the CT scan combined with a simple segmentation (not the one shown here) of the scan achieved with a single, quick thresholding operation with no further processing.

**2.3.2. Creation of the scoliotic model.** To test the procedure, a subject-specific scoliotic model was created from the base model and the data of a specific subject. Virtual palpations were performed on the CT data of a scoliotic subject (56-year-old, female, adult scoliosis, lumbar scoliosis, Cobb angle 75° (Table 2), slice thickness 0.5 mm measured with Surgimap [59,60]).

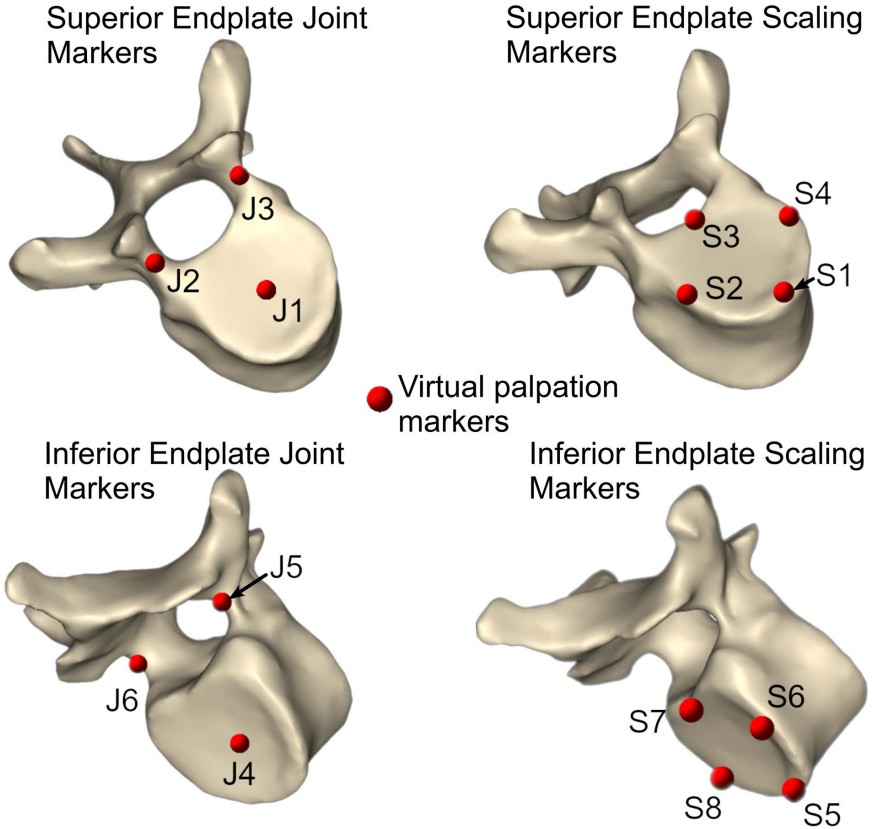

**Fig 3. Anatomical landmark virtual palpations of the vertebrae.** Locations of the virtual palpation landmarks for defining the joints and the scaling of the vertebrae. To define the joints, landmarks are virtually palpated on the top and the base of the pedicle arches and the centre of the superior and inferior endplates. To define the scaling factor, landmarks are virtually palpated on the anterior and posterior-most points and the most lateral points of the superior and inferior endplates.

Following the virtual palpation, a scoliotic model was generated with the OpenSim libraries [18,45] (via a custom script). The workflow for this is as follows (S2 File):

1. The reference systems of the model and the CT data (virtual palpation) were aligned by rotating the virtually palpated anatomical landmarks to correspond with the model reference system (Y axis in OpenSim corresponding to the inferior-superior direction of the CT data) and positioning the markers and the model such that the apex of the sacrum is coincident with the global OpenSim reference system.

2. The model was scaled, specifically, the vertebrae and pelvis were scaled using affine transformations using the Open-Sim scaling tool. Three scaling factors were calculated for each vertebra – corresponding to the anterior-posterior, inferior-superior, and right-left directions. The scaling factors were calculated as the ratio of the Euclidean distance between the existing landmarks on the model and the virtually palpated anatomical landmarks. Specifically, the anterior-posterior scaling factor was calculated using the mean position of the landmarks on the anterior- and posterior-most points of the inferior and superior endplates; the inferior-superior scaling factor was calculated using the mean position of the landmarks on the inferior endplate and the mean position of the markers on the superior endplate; the right-left scaling factor was calculated using the mean position of the landmarks on the right- and left-most points of the inferior and superior endplates. Their masses were scaled based on the total mass of the subject (subject-specific mass of 75 kg

**Table 2. Radiographic evaluation of the standing X-ray of the scoliotic case used in the study using Lenke Classification [61]. Third curve parameters are not reported as the Cobb angle for the third curve was <10°. Lumbar lordosis in the supine position is also reported as a substantial difference between supine and standing was seen in this parameter. All parameters were measured with Surgimap.**

| Radiographic Evaluation | |
|---|---|
| Pelvic tilt | 18.2° |
| Pelvic incidence | 45.9° |
| Sacral slope | 27.7° |
| Lumbar lordosis – standing | 4.3° |
| Lumbar lordosis – supine | 34.0° |
| Thoracic kyphosis | 11.0° |
| Pelvic incidence-lumbar lordosis mismatch | 44.6° |
| Main curve Cobb angle | 75° |
| Main curve type | Lumbar |
| Main curve apex | L2 – L3 |
| Main curve extent | T12 superior – L4 inferior |
| Secondary curve Cobb angle | 51.9° |
| Secondary curve type | Main thoracic |
| Secondary curve apex | T9-T10 |
| Secondary curve extent | T3 superior – T12 inferior |
| Coronal alignment | −42 mm |
| Sagittal vertical axis | 104 mm |

was assumed). The sacrum was not scaled as an appropriate automatic landmark (apex of sacrum) could not be reliably virtually palpated on the CT scan due to the field of view.

3. The sacrum of the model was aligned to the subject specific sacrum. The sacral slope was sacral slope defined in the CT data as the vector between virtually palpated anatomical landmarks on the posterior- and anterior-most points of the promontory of the sacrum.

4. The joint poses of each IVJ were redefined following the ISB recommendation [57]. For each joint this was implemented by first defining IVJ axes for the scoliotic subject. The right-left axis of the joint as the vector between the anatomical landmarks on the pedicles of the vertebrae and the joint origin (midpoint between the anatomical landmarks on the centre of the endplates immediately inferior and superior to the joint). Following this the anterior-posterior axis of the joint was defined as the average of the lines passing anteriorly through the anatomical landmarks on the midline of the endplates immediately inferior and superior to the joint and on a plane defined by the joint origin and the right-left axis. The inferior-superior axis was then defined as perpendicular to the right-left axis and anterior-posterior axis pointing cephalad. This defined the scoliotic joints in the global reference system of the model. The model joint definitions (origin and orientations) were then updated to correspond to the scoliotic joints.

5. The vertebrae were realigned. Following the definition of each scoliotic IVJ, the vertebrae in the model were realigned based on the vertebral body alignment in the CT data. To define the alignment of the vertebrae a reference system for the body of each vertebra was defined in the global reference system. The inferior-superior axis of the vertebra was defined as the vector between the mean position of the landmarks on the inferior endplate and the mean position of the landmarks on the superior endplate; the anterior-posterior axis was defined as the mean of the vectors defined by the landmarks on the midline of the inferior and superior endplate; the right-left axis was defined as the cross product of

the inferior-superior axis and the anterior-posterior axis. Using this definition the vertebrae were aligned in the corresponding parent joint reference frame.

6. Finally, the bushing forces are realigned with the joint to avoid introducing preloads due to joint stiffness.

An initial evaluation of the method was performed by visual inspection to detect any non-anatomical vertebrae position, orientations, and muscle paths (e.g., muscle-in-bone or bone-in-bone penetration).

## 2.4. Evaluation of inter- and intra-operator variability

Using the previously discussed CT data of a scoliotic subject (56-year-old), the inter- and intra-operator variability was assessed. To assess the inter-operator variability, the palpation was performed following a fixed protocol (S1 File) on the CT scan combined with a simple segmentation of the scan achieved with a single, quick thresholding operation with no further processing. The palpations were performed by five operators (all engineers, three of whom had extensive experience working with the spine in *ex vivo*, *in vivo*, and *in silico* studies), who repeated the procedure three times each (inter-operator dataset). Additionally, to evaluate the intra-operator variability, two of these operators (both with experience working with the spine) performed 10 repetitions (intra-operator dataset). Each marker had anterior-posterior, inferior-superior, right-left coordinates, and a Euclidean position (the Euclidean distance from the origin of the reference system to the marker).

Considering the datasets separately, the displacement of each marker to its corresponding mean position (mean of the marker position from the separate dataset) was calculated in the anterior-posterior, inferior-superior, right-left directions, and as the Euclidean distance. Additionally, the distance in these four directions of each marker to its corresponding overall mean position (overall mean defined as the mean marker position from the combined datasets) was calculated.

The above data were aggregated with two levels of granularity, by using different grouping strategies and calculating the median and inter-quartile range (IQR) values. The more granular level maintained each individual marker, calculating the median and IQR from the aggregated dataset of the repeats for the inter-operator dataset and from the aggregated dataset of the operators for the intra-operator dataset. The less granular level grouped the markers (i.e., ignoring location dependency) together, and the median and IQR from the aggregated dataset of the repeats for the inter-operator dataset and from the aggregated dataset of the operators for the intra-operator dataset were calculated.

All statistical analyses were performed in R (v4.2.1) [62] using the *stats*, *PMCMRplus*, and *irr* packages. The coordinates in the anterior-posterior, inferior-superior, and right-left coordinates, the Euclidean position, and the displacement of the virtual palpation markers were checked for normality using the Shapiro-Wilk test. As the data did not follow a normal distribution (Shapiro-Wilk test $p < 0.05$ for all anterior-posterior, inferior-superior, right-left, and Euclidean displacements in both the inter- and intra-operator datasets), non-parametric tests were used for the statistical tests described later. Using the D'Agostino skewness test [63], the data showed moderate to strong, positive and negative skew depending on the dataset considered (inter- or intra-operator) and the absolute position or displacement analysed (anterior-posterior, inferior-superior, right-left, or Euclidean) (skewness was visible, and the coefficient ranged from −0.81 to 6.86).

Inter- and intra-operator variability was tested with the intraclass correlation coefficient (ICC) and their 95% confidence intervals using a two-way random effects model for absolute agreement for single measurement (ICC 2,1) [64,65]. The inter-operator ICC was calculated using each marker's aggregated absolute position data from the inter-operator dataset. The intra-operator variability was assessed using the aggregated absolute position data for each marker from the intra-operator dataset. The ICC of the markers was assessed by anterior-posterior, inferior-superior, and right-left coordinates as well as their Euclidean position (defined as the Euclidean distance from the origin of the reference system to the anterior-posterior, inferior-superior, and right-left coordinates).

Using the displacements calculated considering the inter- and intra-operator datasets separately, the magnitude of the variation due to the inter- and intra-operator variability was investigated. The mean and standard deviation of the

displacements of each marker in the anterior-posterior, inferior-superior, right-left directions, and as the Euclidean distance was calculated.

The more granular data was analysed with unbalanced Friedman tests to account for dependencies. To test for statistically significant ($p < 0.05$) inter-operator variation, the markers were grouped by operator and for intra-operator variation, they were grouped by repeat. In both cases, the markers were blocked by marker name. As unbalanced Friedman tests were used, in cases of significance post hoc one-to-many two-sided Nemenyi tests were applied to search for trends (such as a specific operator or repetition with significant differences) with a significance level of $p < 0.05$ and a critical z value of 2.728 for the inter-operator variability and 3.164 for the intra-operator variability [66].

Unbalanced Friedman tests were applied to subsets within the more granular level datasets to further analyse the patterns within the datasets. The subsets were organised by operator (for the inter-operator set) and repeat (for the intra-operator set). To analyse the effect of the vertebra level, the vertebra level was defined as the group and the anatomical landmark as the block. Conversely, to investigate the effect of the anatomical landmark, the anatomical landmark was defined as the group and the vertebral level as the block. For a significance level of $p < 0.05$, the post-hoc test had a critical z value of 3.10 vertebra level and 3.35 for anatomical landmark.

Additionally, the displacements from the overall mean position (considering the inter- and intra-operator datasets together) were used to assess the overall variability. The mean and standard deviation of the displacements of each marker in the anterior-posterior, inferior-superior, right-left directions, and as the Euclidean distance was calculated.

## 2.5. Assessment of model accuracy

Models were created for each virtual palpation, and the accuracy of the centres of the vertebral bodies and the curvature was assessed. Additionally, accuracy was assessed with reference to the inter- and intra-operator variability, due to the non-parametric nature of the inter- and intra-operator data.

A custom code in MatLab was used to assess the model's accuracy. The accuracy of the vertebral positions was evaluated with the RMSE between the centres of corresponding vertebral bodies, on the model and the segmented CT data (which was assumed as the ground truth), calculated with respect to the centre of L5. The coordinates of the centres were estimated using the sphere method described above (Fig 1) [56] with the .stls of the scoliotic OpenSim model and the manual segmentation of the CT data (considered the ground truth), relative to the centre of L5.

To investigate the impact of the inter- and intra-operator variability on the accuracy of the resulting spine models, the RMSE of the vertebral centres was analysed with regard to the inter- and intra-operator and overall variability. The effect of the inter-operator variability was considered by analysis of the median and IQR of the vertebral centre RMSEs for the models created from the third palpation performed by each operator. The effect of intra-operator variability on the model anatomical accuracy was considered by analysing the median and IQR of the vertebral centre RMSEs for the models from the two operators who performed the palpations 10 times. The effect of the overall variability was considered by analysing the median and IQR of the vertebral centre RMSEs of all the models.

Furthermore, the impact of the operator variability on the accuracy of the spine model was also assessed based on the curvature of the scoliotic spine. The curvature was calculated by fitting $4^{th}$-order polynomials to the centres of the vertebral bodies, and the match between the model and the segmented CT data was assessed based on the RMSE and $R^2$ values. A $4^{th}$-order polynomial was chosen as it represented the lowest-order polynomial that could be fitted to the centre of the manually segmented vertebrae (Fig 2) with an adjusted $R^2$ value above 0.9 in the front and sagittal planes. The same polynomial order was used for all models to avoid biasing the curvature assessment when comparing spine models.

The Cobb angle (CA) of the main curve, lumbar lordosis (LL), apex of the main curve, and apex of the secondary curve were calculated for all the models. The LL was computed using the sacral slope and the orientation of the body reference frame (which aligned with the vertebral endplates) of L1. The CA was calculated from the orientations of the body reference frames of L4 and T12. Normal distributions for the CA and LL were tested with chi-squared goodness of fit. The

mean, standard deviation, and maximum and minimum values for the models were compared to the measurements from the radiographic evaluation (Table 2). The main curve apex was identified by determining the vertebra with the highest displacement relative to the apex of the sacrum (a common reference system for all models). The main curve apex of the model was then compared to the main curve apex identified in the radiographic evaluation (Table 2).

The CT scans only extended from the sacrum to T9. Additionally, the acetabulum and iliac crest were not completely visible. Therefore, secondary curve characteristics, pelvic parameters, or sagittal and coronal alignment could not be included in evaluating the overall anatomical accuracy of the model.

## 3. Results

### 3.1. Creation of a scoliotic spine model

The code successfully created a personalised scoliotic musculoskeletal model from the base spine model, and a virtually palpated landmark set (Fig 4). The vertebrae were scaled, the curvature of the spine was adequately adjusted, and the muscles wrapped meaningfully around the vertebral bodies. The visual evaluation of the model reconstruction implied that the reconstruction method was effective for the majority of the models.

### 3.2. Inter- and intra-operator variability

Interclass correlation coefficient (two-way random effects, single, with agreement model) showed excellent inter- and intra-operator repeatability (>0.9) for all variability parameters (inter- and intra-operator in the anterior-posterior, inferior-superior, right-left, and Euclidean distance for all the markers (Fig 3)).

At the less granular level, the maximum mean displacement of the markers from their mean positions is 2.06 mm; however, the standard deviation is always larger (maximum 3.15 mm) for both the inter- and intra-operator variability (Table 3). The Friedman tests provided a more detailed statistical analysis.

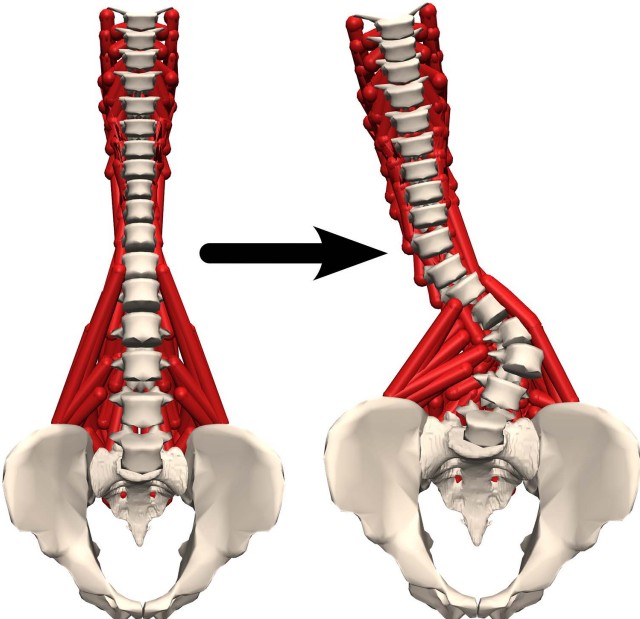

**Fig 4. Creation of the scoliotic spine model.** The base spine model and the resulting scoliotic model, after applying the workflow. The muscles can be seen wrapping around the vertebral bodies.

**Table 3. Mean (standard deviation) displacement from the mean position of the markers for the inter-operator and intra-operator datasets.**

**Inter-operator Variability**

| Opera-tor | Anterior-Posterior displacement, mm | Inferior-Superior displacement, mm | Right-Left displacement, mm | Euclidean displacement, mm |
|---|---|---|---|---|
| 1 | 1.00 (1.23) | 1.23 (1.27) | 0.64 (0.84) | 1.95 (1.72) |
| 2 | 0.93 (1.28) | 1.12 (1.11) | 0.62 (0.91) | 1.85 (1.66) |
| 3 | 1.02 (1.78) | 1.12 (1.12) | 0.72 (0.85) | 1.94 (2.05) |
| 4 | 1.07 (2.59) | 1.05 (1.16) | 0.73 (1.68) | 1.93 (3.15) |
| 5 | 1.10 (2.14) | 1.25 (1.43) | 0.67 (1.09) | 2.06 (2.61) |

**Intra-operator Variability**

| Repeat | Anterior-Posterior displacement, mm | Inferior-Superior displacement, mm | Right-Left displacement, mm | Euclidean displacement, mm |
|---|---|---|---|---|
| 1 | 0.85 (0.79) | 1.15 (1.19) | 0.54 (0.63) | 1.74 (1.33) |
| 2 | 0.47 (0.47) | 0.64 (0.69) | 0.30 (0.29) | 0.96 (0.76) |
| 3 | 0.72 (0.77) | 0.75 (0.86) | 0.44 (0.55) | 1.31 (1.10) |
| 4 | 0.62 (0.60) | 0.71 (0.75) | 0.36 (0.35) | 1.14 (0.88) |
| 5 | 0.62 (0.72) | 0.64 (0.72) | 0.36 (0.46) | 1.09 (0.99) |
| 6 | 0.62 (0.62) | 0.71 (0.76) | 0.38 (0.50) | 1.16 (0.95) |
| 7 | 0.51 (0.52) | 0.65 (0.79) | 0.30 (0.35) | 1.01 (0.89) |
| 8 | 0.54 (0.69) | 0.66 (0.73) | 0.34 (0.78) | 1.05 (1.17) |
| 9 | 0.49 (0.68) | 0.61 (0.73) | 0.31 (0.58) | 0.96 (1.06) |
| 10 | 0.55 (0.59) | 0.71 (0.83) | 0.38 (0.44) | 1.10 (0.99) |

For the inter-operator variability, the Friedman test showed significant differences ($p < 0.05$) in the position of the marker placement in the inferior-superior direction. The post-hoc Nemenyi test showed that there were significant differences between operator 1 and operators 3, 4, and 5 but not between operator 1 and 2.

For the intra-operator variability, the Friedman test showed significant differences ($p < 0.05$) in the position of the marker placement in the anterior-posterior, inferior-superior, right-left, and Euclidean displacement. However, the post-hoc Nemenyi test did not calculate significant differences when performing the one-to-many test in the anterior-posterior or right-left displacement. In the inferior-superior displacement, the post-hoc test detected significant differences between repeats 1 and 2, 5, 4, 8, 9, and in the Euclidean displacement, significant differences were found between repeat 1 and all other repeats.

Although there were significant differences in the marker displacement due to vertebra level within the inter-operator dataset, no clear trends of one vertebra always causing a significant difference or a significant difference consistently in one direction were present (Table 4). However, when the anatomical landmarks were analysed, a clear trend of significant differences in the inferior-superior direction became apparent (Table 4).

Similar patterns were identified from the same analysis performed on the intra-operator dataset. Significant differences did not seem to be associated with a specific vertebral level or direction when grouping by vertebra. Significant differences were present when grouping by anatomical landmark, and although they did not seem to be associated with a particular landmark, most significant differences occurred in the inferior-superior direction.

The overall variability of the markers was calculated based on the displacement of the markers from the mean marker position across all repetitions and operators in the anterior-posterior, inferior-superior, right-left, and Euclidean positions (Table 5). The mean variation of the displacement was under 2 mm with a standard deviation of 2.06 mm.

Analysis of the IQR of the absolute position of the markers found that outliers were present within the IQR up to a maximum of 10.3 mm (Fig 5). The mean variability did not exceed 2 mm.

**Table 4. Results of the Friedman and post-hoc Nemenyi test conducted on the dataset for each operator.** Testing for significant differences due to vertebra level and anatomical landmark (anatomical landmark defined by marker location, Fig 3). Significance code reported for the post-hoc Nemenyi test: *** [0, 0.001], ** [0.001, 0.01], * [0.05, 0.1],. [0.1, 1].

| Operator | Significant effect of vertebra level | | | | | | | |
|---|---|---|---|---|---|---|---|---|
| | Anterior-Posterior displacement, mm | | Inferior-Superior displacement, mm | | Right-Left displacement, mm | | Euclidean displacement, mm | |
| | Friedman | Post-hoc | Friedman | Post-hoc | Friedman | Posthoc | Friedman | Post-hoc |
| 1 | No | | Yes | ** L1-L2 . L1-T10 | No | | Yes | |
| 2 | Yes | | No | | No | . L1-L2 | Yes | ** L1-T9 |
| 3 | No | | No | . L1-L3 | No | | Yes | |
| 4 | No | | No | | No | . L1-T12 | Yes | . L1-T10 |
| 5 | Yes | . L1-L3 | Yes | | No | | No | |
| | **Significant effect of anatomical landmark** | | | | | | | |
| 1 | Yes | ** S1-S4 *** S1-S8 | Yes | *** S1-S2 * S1-S3 *** S1-S4 *** S1-S6. S1-S8 ** S1-JM5 * S1-JM6 | Yes | | Yes | . S1-S3. S1-JM2 * S1-JM3 |
| 2 | Yes | | Yes | ** S1-S3 * S1-S7 | Yes | | Yes | . S1-JM1 * S1-JM2. S1-JM4 |
| 3 | Yes | * S1-S4 | Yes | * S1-S2 * S1-S3 *** S1-S4 ** S1-S6 * S1-S7 *** S1-S8 ** S1-JM1. S1-JM3 ** S1-JM4 | Yes | | Yes | |
| 4 | Yes | * S1-JM2 | Yes | * S1-JM5 * S1-JM6 | Yes | * S1-S6 | Yes | |
| 5 | Yes | . S1-JM6 | Yes | ** S1-S3 ** S1-S7 * S1-JM1. S1-JM4. S1-JM6 | Yes | | Yes | . S1-JM1. S1-JM3 * S1-JM4 |

### 3.3. Model accuracy

The centres of the vertebral bodies were successfully found for both the segmentation of the CT data and the model (Fig 6) and 4th-order polynomials were fitted in both the sagittal and frontal planes (Fig 7). A visual inspection of the positions of the vertebral centres showed that the positions were generally consistent between models with limited variance, however there was an offset from the CT vertebral centres at many of the vertebral levels (Fig 6) indicating a component of systematic error in addition to the random error. This was consistent with the errors of the vertebral centres and polynomial curves (Tables 6 and 7).

The median errors represent the systematic errors and the IQR errors represent the random errors which can be associated with operator variability. The overall IQR is 14% of the median RMSE in the Euclidean direction, the variability of the curvature accuracy is also 14–16% of the median error (Table 6). Analysing this based on the absolute error, the variability of the accuracy of the models (the IQR) (Table 7) is on average: 33% of the median absolute error in the anterior-posterior

**Table 5. Mean variation of the marker positions compared to the overall mean position of the markers.** Overall variability of the markers in the anterior-posterior, inferior-superior, right-left positions, and Euclidean positions described by the mean and standard deviation of the interquartile ranges for all the markers.

| Position | Displacement | |
|---|---|---|
| | Mean, mm | Standard deviation, mm |
| Anterior-Posterior | 0.95 | 1.60 |
| Inferior-Superior | 1.19 | 1.23 |
| Right-Left | 0.61 | 0.99 |
| Euclidean | 1.88 | 2.06 |

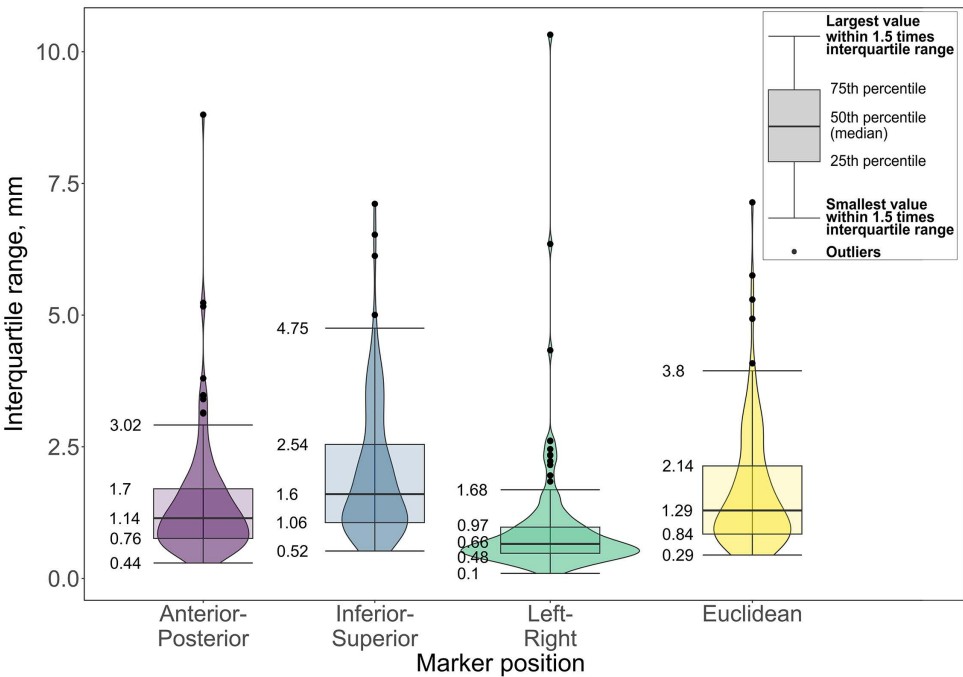

**Fig 5. Distribution of the variation of the virtually palpated anatomical landmarks.** Variation of the interquartile range of the absolute position showing the median, interquartile range, and outliers of the IQR of the absolute position of the markers in anterior-posterior, inferior-superior, right-left positions and Euclidean positions. The violin plots show the distribution of the data.

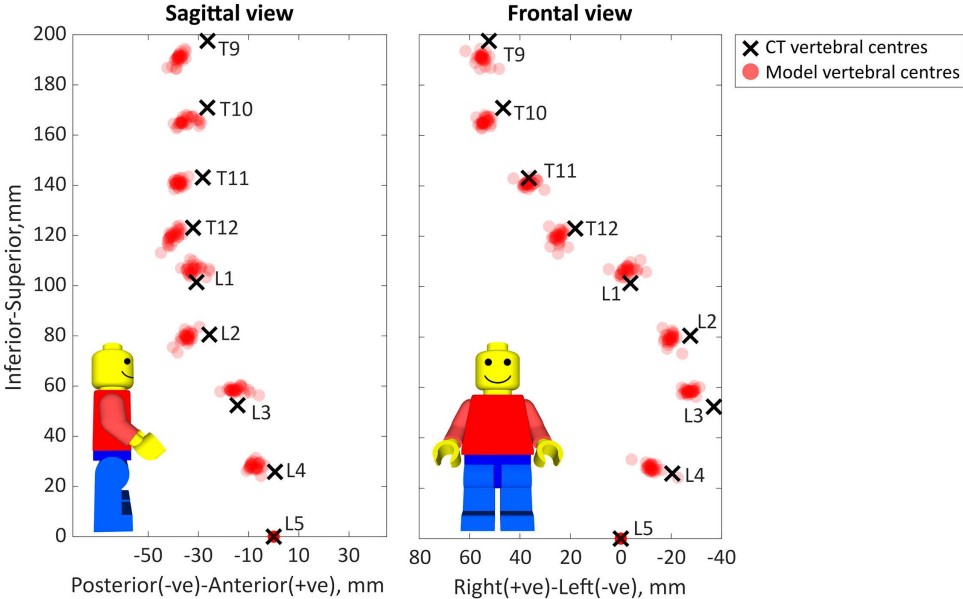

**Fig 6. The centres of the vertebrae from the models and the segmented CT data in the sagittal and frontal views.** The bold X shows the location of the vertebrae centres from the ground truth CT scan, the centre of the vertebrae for each model are represented by semi-transparent red circles, darker red indicates many of the models had the vertebrae at that specific level in a similar location.

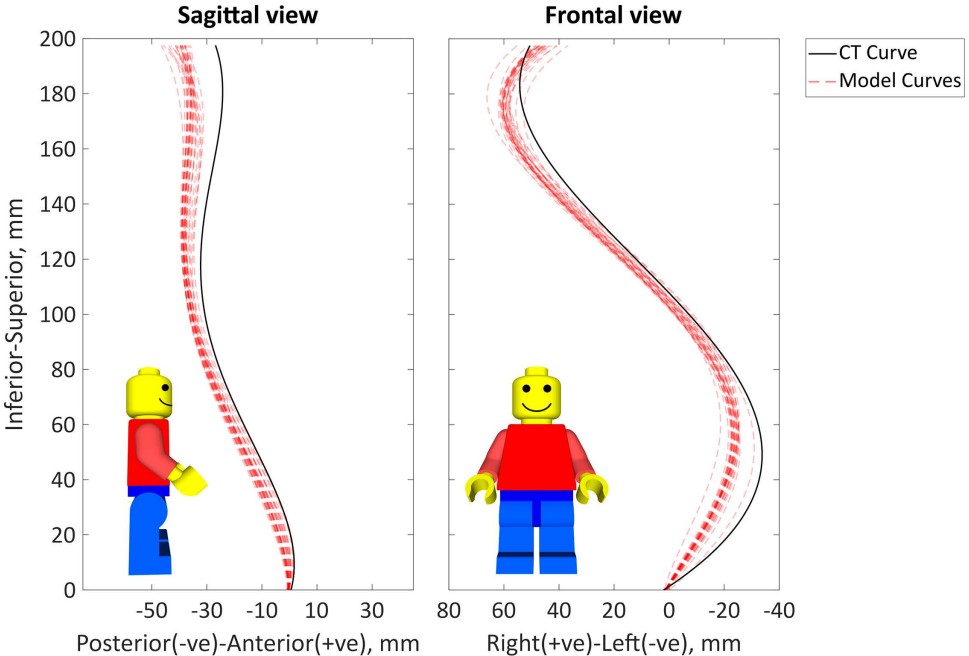

**Fig 7. The 4th order polynomial line fitted to the vertebral centres of the segmented CT data and the models.** The solid black line shows 4th order polynomial line fitted to the centres of the ground truth segmented CT data, the 4th order polynomial line fitted to the centres of the vertebrae for each model are represented by semi-transparent red dashed line, darker red indicates many of the models had a similar 4th order polynomial line fitted.

direction, 41% of the median absolute error in the inferior-superior direction, 35% of the median absolute error in the right left direction, and 22% of the median absolute Euclidean distance.

The median and IQR errors were similar for intra-, inter-, and overall variability (Table 6, Figs 6 and 7). Thus suggesting the variability of the accuracy of the reconstructions was not associated specifically with inter- or intra-operator variability. The RMSE for the curves in the sagittal and frontal planes were similar, indicating the mean error in one plane was not greater than in the other. However, the $R^2$ value was notably better in the frontal plane than the sagittal plane, indicating most of the systematic error may occur in the sagittal plane (Table 6, Figs 6 and 7). This is further supported as the median errors of the vertebral centres were larger in the anterior-posterior direction than in the right-left direction, while the IQR of the errors were similar (Table 7).

The maximum systematic error occurred at the T9 for the majority of the models, with the largest errors occurring in the sagittal plane (Table 7, Figs 6 and 7). A notable exception to this is the largest systematic error occurred at L3 in the right-left direction which corresponded the apex of the main curve was at L2-L3 (Table 2) and the most severely deformed vertebra (Fig 3).

The Cobb angles and lumbar lordosis were normally distributed. The mean Cobb angle was in close agreement with the radiographic evaluation, whereas there were substantial systematic errors (>20°) at the maximum and minimum angles (Table 8). The mean lumbar lordosis agreed well with the radiographic evaluation of the supine radiographs but poorly with the standing radiographs. The maximum lumbar lordosis was within the range of the supine radiograph, while the systematic error of the minimum lumbar lordosis was more substantial. The standard deviation for both Cobb angle and lumbar lordosis was found to be within or in close proximity to the uncertainty of radiographic measurements. The apex of the curve was found at L2 for all models.

## 4. Discussion

This study aimed to create a codified workflow to create a severely scoliotic subject-specific musculoskeletal model from a generic spine model and a set of virtually palpated landmarks on the subject-specific CT data. A generic model was developed

**Table 6. The variability of the reconstruction accuracy due to the intra-, inter-, and overall operator variability.** The median and interquartile range of the Euclidean root mean square error of all the vertebral centres, the RMSE of the fitted curves in the sagittal and frontal planes, and the $R^2$ of the fitted curves in the sagittal and frontal planes.

| Variability factor | Median (and IQR) vertebral centre Euclidean RMSE, mm | Median (and IQR) curve RMSE, mm | | Median (and IQR) curve $R^2$ | |
|---|---|---|---|---|---|
| | | Sagittal | Frontal | Sagittal | Frontal |
| Inter-operator | 10.0 (1.9) | 5.9 (2.5) | 6.6 (0.5) | 0.73 (0.2) | 0.96 (0.0) |
| Mean Intra-operator | 11.0 (1.0) | 7.0 (0.9) | 6.9 (1.1) | 0.63 (0.1) | 0.96 (0.0) |
| Overall | 11.1 (1.5) | 7.0 (1.1) | 7.1 (1.0) | 0.63 (0.1) | 0.95 (0.0) |

**Table 7. The median and interquartile range of the absolute error across all the models at each vertebral level in the anterior-posterior (AP), inferior-superior (IS), right-left (RL) directions, and the Euclidean distance. The average median, interquartile range, and the RMSE of the errors in each direction across all levels grouped together are reported. The final three rows report the mean, standard deviation, and RMSE of the absolute errors of the reconstruction performed by the manual segmentation in the study by Overbergh et al. [24].**

| Vertebral level | | Median absolute error (IQR), mm | | | |
|---|---|---|---|---|---|
| | | Anterior-Posterior | Inferior-Superior | Right-Left | Euclidean |
| T9 | | 11.41 (1.77) | 6.82 (1.58) | 2.92 (1.67) | 13.37 (2.10) |
| T10 | | 9.59 (4.23) | 5.85 (1.78) | 7.61 (2.05) | 13.55 (5.03) |
| T11 | | 9.45 (1.92) | 2.25 (1.44) | 1.36 (1.67) | 9.81 (1.69) |
| T12 | | 8.07 (3.55) | 3.31 (2.16) | 6.52 (2.01) | 11.64 (4.00) |
| L1 | | 2.01 (2.09) | 5.07 (1.99) | 2.45 (2.51) | 6.25 (1.40) |
| L2 | | 8.62 (1.56) | 1.41 (1.60) | 7.81 (2.15) | 12.25 (1.47) |
| L3 | | 2.08 (1.79) | 6.13 (0.89) | 9.65 (2.42) | 11.63 (1.55) |
| L4 | | 8.20 (2.82) | 2.06 (1.95) | 8.49 (1.77) | 12.13 (2.46) |
| Median | | 7.43 | 4.11 | 5.85 | 11.32 |
| IQR | | 2.47 | 1.67 | 2.03 | 2.46 |
| RMSE | | 7.74 | 4.54 | 6.36 | 11.00 |
| Overbergh et al. [24] | Mean | 0.83 | 0.71 | 1.59 | 2.09 |
| | SD | 0.67 | 0.59 | 0.83 | 0.89 |
| | RMSE | 1.05 | 0.91 | 1.78 | 2.26 |

**Table 8. The mean (standard deviation), minimum and maximum Cobb angle (CA) and the lumbar lordosis (LL).**

| Spinal curvature measurement | Model | | | Radiograph Evaluation |
|---|---|---|---|---|
| | Mean (SD) | Min | Max | |
| CA, ° | 77 (10) | 59 | 103 | 75 |
| LL, ° | 34 (4) | 19 | 40 | 34 †, 4 * |

† Supine, * Standing.

from an existing spine model [21]. This involved the addition of a new set of landmarks to allow each vertebra to be individually, anisotropically scaled; 6 DoF joints at each vertebral level with a linear spring-damper element to represent the mechanical properties of the IVJ; and wrapping spheres at each vertebral body to ensure the muscle paths do not pass through the vertebral bodies when the scoliotic curvature is introduced. The virtual palpation requires 14 markers per vertebra: 8 to scale the vertebrae and 6 to determine the joint pose. The codified workflow aligned the sacral slope of the model with the CT scan, scaled the vertebrae and then redefined the position and orientation of the IVJs to create the scoliotic curvature.

Users of the workflow only need to provide the set of virtually palpated anatomical landmarks (S1 File) and run the code to a create scoliotic spine models (S2 File). The code also allows for further modifications such as subject mass and subject specific IVJ stiffness. All the necessary codes and files are available on the OpenSim website (Personalised scoliotic spine models) and on FigShare (https://doi.org/10.6084/m9.figshare.28912175). The workflow does not require the user to provide segmentations of the vertebrae, rather a statistical shape model of the sacrum and the original .vtp files of the vertebrae are used. The .vtp files of the vertebrae suffer from low resolution. Therefore, while the scaling may result in the model adequately represent the geometry of healthy vertebrae, in cases where the vertebrae are severely deformed the scaled model may not represent the deformed vertebra shape accurately. Furthermore, the code does not apply subject specific mass distribution or apply different stiffnesses at different IVJ levels. Future users could modify of the code to allow for these properties as well – a function is already provided to easily edit the IVJ stiffness, the desired stiffness simply needs to be specified.

The analysis of the operator variability was based on the positions of the virtually palpated anatomical landmarks. The intraclass correlation coefficient showed that the virtual landmark palpation positions could be selected with excellent reliability between operators and between repeats for the same operator. Nonetheless, the inter- and intra-operator variability resulted in significant differences of the marker placement, predominantly in the inferior-superior direction, implying that identifying the superior and inferior surfaces of the vertebrae is not consistently achieved on medical CT scans. The significant differences were not associated with a specific landmark or vertebra, indicating the protocol is robust. Despite significant differences, the mean magnitude of the differences was under 2 mm (4 slices of the CT data) even in the case of severe scoliosis. This suggests the virtual palpations were robust against operator variability. However, it is necessary to control for outliers.

Both systematic errors and random errors (due to operator variability in the positioning of the virtually palpated anatomical landmarks) affected the accuracy of the models. To assess the contributions of each, the median and IQR of the fitted curves (RMSE and $R^2$) and the RMSE of the Euclidean distance of the centres of the vertebrae (Table 6) and the median and IQR of the absolute error in each direction were analysed (Table 7). The IQR of the RMSE of the vertebral centre and the RMSE and $R^2$ of the curvature suggest that the overall model accuracy can be vary by around 15% due to operator variability, with larger variability (30–40%) seen when examining the absolute error in the individual directions. This implies the inter- and intra-operator variation affected the model accuracy less than the systematic errors. The difference between the errors due to the inter-operator variation and the errors due to intra-operator variation were negligible compared to the overall errors. Future work should focus on improving the accuracy of the model, this may focus on improving identification of the vertebra centre and the vertebral endplate centre. Furthermore, errors due to operator variability could be removed by automatically identifying the anatomical landmarks.

As was the primary aim of the current study, the workflow allows for the rapid generation (15–30 minutes per vertebra to be included in the model) of a scoliotic MSK model. This was achieved by eliminating the need for the time-consuming process of manual segmentation which the current gold-standard models require [24], a process which the present study found requires approximately an hour per vertebra. However, the time-savings come with a modest reduction of accuracy compared to the gold-standard model. The gold-standard modelling approach from Overbergh et. al. 2020 [24] achieved mean errors of 0.71–1.59 mm depending on the direction and 2.09 mm in the Euclidian direction while the approach in the current study had median errors between 4.11–7.43 mm depending on direction and 11.32 mm in the Euclidian direction. There are multiple factors that may contribute to these systematic errors. The use of generic vertebra geometries rather than subject-specific geometries. The geometries can differ substantially, especially at levels where severe deformation is present, such as at L2 and L3 in the case of the subject used in the current study (Fig 2). This would reduce the correspondence of the positioning of the virtual palpitations between the model and the CT scan. This would alter the scaling and joint definition and would affect the position of the vertebral centres. One such example of this can be seen in the mismatch between vertebrae L2 to L5 (Fig 6) where the deformation of the L3 vertebra reduced the clarity of the L2

inferior endplate and the L4 superior endplate in the CT scan. Other studies which have used similar methods to construct scoliotic models with generic vertebral geometries have not reported the accuracy of the position of the centre of each vertebra [13,23]. Therefore, it is challenging to assess the influence of a generic compared to subject-specific geometry on the accuracy of the vertebral position. Nonetheless, the errors in the present study were large (as median errors up to 11.3 mm). Future work should focus on reducing this error.

There was a strong agreement in the shape of the curvature of the models and the curvature estimated from the CT scan in the frontal plane, whereas the agreement was weaker in the sagittal plane. The mismatch of the curvature in the frontal plane was due to an overextension of the models. The flexion-extension of the joints was defined by the joint markers on the centre of the endplates. This suggests a better definition or method is required to identify the vertebral endplate's centre. It should also be considered that the definition of the joint orientation was based on a non-pathological spine, meaning this definition may not be suitable for severe scoliosis.

Despite the weaker agreement in curve shape in the sagittal plane, the RMSE of the sagittal and frontal curves were similar. However, analysis of the RMSE of the vertebral centres in the anterior-posterior, inferior-superior, and right-left directions indicated higher systematic errors in the anterior-posterior direction, corresponding to sagittal plane errors. The errors in the anterior-posterior direction also increased in the cranial direction, which can be explained by the overextension of the models.

The Cobb angle of the model agreed well with the same angle evaluated on the original CT scan, and the standard deviation was within the same order of magnitude as the uncertainty associated with radiographic evaluations, which is commonly stated to be 5° [67,68]. The apex of the curve for the model was found at L2 for all models. This agreed well with the radiographic evaluation, which reported the apex of the curve to be at L2-L3. The model lumbar lordosis matched the lumbar lordosis in the supine position. However, there was a large systematic error compared to the standing position. Differences in the lumbar lordosis are expected when comparing the standing and supine positions [69]. The CT was acquired in the supine position, explaining the concordance between the model and the lumbar lordosis in the supine position. The standard deviation of the lumbar lordosis was even lower than that of the Cobb angle, suggesting the model was able to capture these two parameters accurately. This suggests that the workflow can create models that could be used for simulation of surgical procedures which are performed with the patient in the prone position. However, it presents a challenge for post-operative simulations when the subject may be upright and for simulations used to estimate the IVJ stiffness – which often use data from experiments that require the patient to be standing [70]. The sacral slope was not evaluated, as the alignment of the model sacral slope to the sacral slope of the CT scan was enforced, as this was used to define a common reference system for the model and the CT scan.

Several other limitations to this study should be considered. Only one subject was used in this study, and future studies should increase the number of participants. However, the subject used represents a severe case of adult scoliosis (severe scoliosis defined as Cobb angle >50° [71]), possibly one of the more complex types of scoliosis to model, as degeneration and compensatory mechanisms may be present. Therefore, adapting the methodology to represent less severe or complex cases should be possible in future work.

Spinopelvic parameters (such as pelvic incidence, pelvic tilt, thoracic kyphosis, and lumbar lordosis mismatch – the difference between the pelvic incidence and the lumbar lordosis [72]) and sagittal balance are important parameters in surgical planning and decision making and influence the outcome of surgical corrections [72–76]. Poor management of these parameters has been associated with poor post-operative outcomes, such as post-operative lower back pain [72], proximal junctional kyphosis [73,74], distal junction kyphosis [74], disk degeneration [74], mechanical complications (instrumentation failure) [75], compensatory mechanisms to maintain balance (resulting in greater energy expenditure) [75], and generally lower health-related quality of life [76]. Thus, evaluation of a wide range of spinopelvic parameters and spinal alignment is important. Unfortunately, the CT scan only extended to T9, and the field of view did not capture the entire pelvis. Likewise, an X-ray with the femoral heads extending to the cervical spine was

unavailable meaning a more comprehensive comparison of spinopelvic parameters to the radiographic evaluation was not possible.

Furthermore, the accuracy of the muscle paths was only evaluated with a visual inspection in terms of anatomical plausibility (no muscle-in-bone penetration). Assessment of the patient-specific muscle path accuracy was not possible as magnetic resonance imaging (MRI) data was not available to establish the ground truth of the muscle paths. Indirect validation of the muscle paths based on the on parameters (such as joint moments and contact forces) from simulation predictions [77] was not possible as functional test data was not available.

Future studies should include an evaluation of spinopelvic parameters and CT or MRI scans to assess the muscle paths. The model incorporated generic stiffnesses; an accurate representation of the scoliotic spine would require subject-specific stiffnesses, and it is expected that reuse of this model will incorporate that aspect. To this end, the package provided to create the subject-specific models includes an easy-to-use function which allows the user to assign the desired stiffnesses. As part of this, methodologies will need to be found to move the model from a supine pose to an upright pose. One possible approach would be to perform a registration between the CT data (typically taken in a supine position) and a bi-planar X-ray (typically taken in a standing position) to calculate an individual transformation matrix for each vertebra to move between the two poses.

There are several key topics that should be the subject of future studies before models from this workflow is used in clinical scenarios, surgical simulations or for optimisation of implant design. The effect of the palpations (and thus the vertebra positions) on simulation outcomes and the effect of the muscle on the necessary surgical corrective forces should be investigated; additionally, methods to tune the muscle properties should be established to ensure physiological behaviour.

In conclusion, the codified workflow (OpenSim project: Personalised scoliotic spine models) generates subject-specific models which accurately capture the Cobb angle and lumbar lordosis. This is the first semi-automatic workflow to create scoliotic models that include muscle paths that wrap around the vertebrae. The models appear to be robust to inter- and intra-operator variability. However, further work is needed to improve the definition of the intervertebral joint poses, likely reducing errors in the vertebral positions.

## Supporting information

**S1 File. Virtual Palpation Protocol.** Protocol for performing the virtual palpation of the anatomical landmarks and the accompanying naming protocol for the landmarks.
(PDF)

**S2 File. Generation of scoliotic spine.** Document describing in detail the codes that are provided as part of the workflow and how to use them.
(PDF)

## Acknowledgments

The authors would like to thank Dr. Jennifer Fayad, Dr. Chloe Techens, Dr. Francesca Bottin, and MEng Alex Bersani for performing the virtual palpations used to analyse the inter- and intra-operator variability.

## Author contributions

**Conceptualization:** Samuele Luca Gould, Marco Viceconti.

**Data curation:** Samuele Luca Gould.

**Formal analysis:** Samuele Luca Gould, Giorgio Davico, Monica Cosentino.

**Funding acquisition:** Luca Cristofolini.

**Investigation:** Samuele Luca Gould, Giorgio Davico.

**Methodology:** Samuele Luca Gould, Giorgio Davico, Monica Cosentino.

**Project administration:** Samuele Luca Gould.

**Software:** Samuele Luca Gould.

**Supervision:** Giorgio Davico, Luca Cristofolini, Marco Viceconti.

**Visualization:** Samuele Luca Gould.

**Writing – original draft:** Samuele Luca Gould.

**Writing – review & editing:** Samuele Luca Gould, Giorgio Davico, Monica Cosentino, Luca Cristofolini, Marco Viceconti.

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
