## [Decision Letter · Decision Letter 0]

7 Jul 2025

Dear Dr. Gould,

In particular, the reviewers have pointed out the need of providing more details about the methodology and improving the discussion of the results.

We look forward to receiving your revised manuscript.

Kind regards,

Giulia Pascoletti, Ph.D.

Academic Editor

PLOS ONE

 [HEU H2022 project “Metastra - Computer- Aided Effective Fracture Risk Stratification Of Patients With Vertebral Metastases For Personalised Treatment Through Robust Computational Models Validated In Clinical Settings” (topic HLTH-2022-12-01, grant ID 101080135).

Reviewers' comments:

Reviewer's Responses to Questions

**Comments to the Author**

1. Is the manuscript technically sound, and do the data support the conclusions?

Reviewer #1: Yes

Reviewer #2: Partly

2. Has the statistical analysis been performed appropriately and rigorously?

Reviewer #1: Yes

Reviewer #2: No

3. Have the authors made all data underlying the findings in their manuscript fully available?

Reviewer #1: Yes

Reviewer #2: Yes

4. Is the manuscript presented in an intelligible fashion and written in standard English?

Reviewer #1: Yes

Reviewer #2: Yes

Reviewer #1: The manuscript titled “Generation of severly scoliotic subject-specific musculoskeletal models” presents a semi-automatic workflow for the generation of subject-specific musculoskeletal models of the scoliotic spine, with a focus on severe deformities. The study is timely and relevant, and the effort to build a reproducible and open-source tool is truly commendable. The manuscript falls within the scope of Plos One journal and represents a relevant study in the advancement of musculoskeletal spinal biomechanics state-of-the-art. However, there are some points that require further clarification or improvement, below described:

- The description of how the vertebrae were displaced relative to one another during the reconstruction process still requires improvements. The authors mention aligning the vertebrae based on virtually palpated anatomical landmarks and applying affine transformations, but the specific procedure used to reposition the vertebral bodies to achieve the target curvature lacks sufficient detail. Similarly, the method for anisotropic scaling of the vertebrae is not fully explained. Although affine transformations are said to be computed using specific landmarks, a more precise and step-by-step description would significantly improve the manuscript.

- The definition of the intervertebral joint based on ISB recommendations is appropriate; however, the implementation itself would benefit from a brief explanation. Additionally, the manuscript mentions the addition of linear and spring dampers in the form of bushings, but several critical aspects are not clarified: were the connectors inserted after the morphological adjustments? Otherwise a pre-load would be introduced due to the roto-translation of adjacent vertebrae following the new curvature. Were the bushings replaced in the same relative location of traditional spherical joints?

- Concerning the virtual palpation performed by the operators, it is unclear whether they were clinicians or engineers. This distinction is important, as it relates directly to the reliability of the process. Furthermore, it is essential to specify what instructions were given to the operators for the virtual palpation. Although the variability analyses are detailed and statistically thorough, without knowledge of the protocol followed by the operators, it is difficult to assess whether the observed reliability is due to the method itself or to the operators’ expertise.

- Model accuracy is primarily assessed based on vertebral center position and curve fitting. While this is a valuable metric, but how anatomically implausible results (e.g., muscle paths intersecting bone) were identified and addressed? Was this done purely via visual inspection, or was a systematic metric or validation method used?

- A noticeable mismatch is observed between vertebrae L3 to L5 in Fig. 6. The lumbar vertebrae appear to be more spaced than expected, especially imagining the posterior region where the facet joints are located. It would be important to clarify whether this discrepancy is due to the scaling procedure, landmark definition, or reference alignment. Was this a visual artifact? How the authors think this could reflect a recurring issue in the model output?

- In the results, lumbar lordosis is reported as a negative angle. While this is likely due to an inverted curvature, angle convention generally refer to positive angles. Moreover, in the table, some terminology doesn’t appear straightforwardly clear: what is intended for “main curve type” and “main secondary curve type” ? Do they refer to anatomical regions (e.g., thoracic vs. lumbar)? Similarly, the term “pelvic mismatch” is used without adequate context regarding its clinical significance or its impact on the model.

- The authors point out that the reconstructed models align well with CT scans acquired in the supine position but not with standing radiographs. This could turn into significant limitation of musculoskeletal models based on CT data. Since evaluations and model-based analyses are typically conducted in standing posture, the inability of the model to replicate anatomical parameters across postures could impact the reliability of the estimations. This limitation should be emphasized more strongly in the Discussion for further debate with scientific community.

- The statement on page 25 suggesting that systematic error affected the model more than operator variability appears in contradiction with earlier results where inter-operator differences were significant in several directions. Could the authors please motivate the sentence?

- The issue of low reproducibility in vertebral geometry is mentioned, which is a known challenge in musculoskeletal modeling. In this perspective, does the proposed opensource tool requires a prior adjustment of vertebral geometry, or this is handled automatically by the workflow, or the method can also be applied to traditional vertebral .vtp files (which often suffer from low resolution)? The authors should consider adding a summary table or schematic—either in the Discussion or as Supplementary Material—outlining the expected inputs required for the tool shared opensource, which are the key steps, and the limitations of the current pipeline. This would greatly enhance the practical utility of the work.

Reviewer #2: The introduction is largely clear and poses an interesting research aim. The promise of an open-source package to generate subject-specific MSK spine models from CT is very welcome.

Unfortunately, I believe that the manuscript lacks many relevant details and that the proposed code requires further development to achieve accuracy relevant for publication. Please see my more specific comments below:

1. First, regarding the accuracy of the model, in the results a comparison is made with another paper where submillimeter accuracy was achieved, whereas the current proposed methodology leads to errors of multiple millimeters. Considering that this model was developed and tested on only one subject leads me to seriously doubt the reliability of the model. The authors rightfully mention in the discussion that 'a better definition or method is required to identify the vertebral endplate's center'.

2. A key point of the methodology is the virtual palpation of CT scans. The protocol for how to do this is missing. It is also not fully clear to me if the palpations are actually performed on the CT scans, or if they are performed on segmentations of the spine. If segmentation of the spine is required before palpations can be done, I do not believe there is much merit in this method over directly registering the vertebrae of the generic model to the segmentations.

3. In relation to the previous comment, it is unclear how the operators were instructed to perform the palpations and what their experience level is working with CT images (were these operators radiologists, orthopedic surgeons, students?).

4. A very big portion of the manuscript is about the interobserver and intraobserver variation. I believe this to be of minor relevance to the actual accuracy of the model. If instructions for virtual palpations are good enough to reach submillimeter repeatability, then that's sufficient. I suggest reporting the intraclass correlation coefficients and put more emphasis on the actual accuracy of the model. Especially the information about all repetitions (table 3) and comparisons between individual operators (table 4) doesn't mean much in my opinion.

5. The aim posed in the introduction strongly suggests that muscle-paths and IVJ stiffnesses are personalized. The IVJ stiffnesses are not personalized, and the muscle paths rely on the bone shape and predefined connection points of a generic healthy spine. The only true subject-specificity is the position and size of the vertebrae, the rest naturally follows from how the generic model was already developed and is thus not novel for this study.

6. The effect of the palpations on the final positions of the vertebrae in the model is well-presented. It would also be interesting to see how this would affect actual modelling outcomes. Is the difference in position relevant for intended use of this model? The introduction mentions that these model can be used to predict treatment outcomes and simulate instrumentation strategies. It would be interesting to see some examples tested in this paper. Similarly, what is the effect of having the muscles go around the vertebrae on model predictions? And, what is the effect of adding the translational DoFs to the IVJ?

7. The introduction states '...development of high-quality automatic segmentation methods, as the review by Galbusera et al. notes, work is ongoing (26).'. The reference here is 6 years old and plenty of development has been completed since then. Also already 5 years ago open-source neural networks have been published for automatic segmentation of the spine from CT, e.g.,:

Payer, C., Štern, D., Bischof, H., Urschler, M., 2020. Coarse to Fine Vertebrae Localization and Segmentation with SpatialConfiguration-Net and U-Net:, in: Proceedings of the 15th International Joint Conference on Computer Vision, Imaging and Computer Graphics Theory and Applications. Presented at the 15th International Conference on Computer Vision Theory and Applications, SCITEPRESS - Science and Technology Publications, Valletta, Malta, pp. 124–133. https://doi.org/10.5220/0008975201240133

8. More details need to be mentioned regarding the patient and the scans under 2.1. Apparently there is also an x-ray that now only gets mentioned in table 2.

9. For future reviews: please add line numbers to make referencing text easier.

**Do you want your identity to be public for this peer review?** For information about this choice, including consent withdrawal, please see our Privacy Policy

Reviewer #1: No

Reviewer #2: **Yes: ** Joeri Kok

---

## [Author Response · Author response to Decision Letter 1]

18 Aug 2025

Reply to Reviewer #1

Reviewer #1: The manuscript titled “Generation of severly scoliotic subject-specific musculoskeletal models” presents a semi-automatic workflow for the generation of subject-specific musculoskeletal models of the scoliotic spine, with a focus on severe deformities. The study is timely and relevant, and the effort to build a reproducible and open-source tool is truly commendable. The manuscript falls within the scope of Plos One journal and represents a relevant study in the advancement of musculoskeletal spinal biomechanics state-of-the-art. However, there are some points that require further clarification or improvement, below described:

Authors: The authors wish to thank the Reviewer for having read the paper, the positive comments, and the detailed recommendations for improving the paper.

Reviewer #1: The description of how the vertebrae were displaced relative to one another during the reconstruction process still requires improvements. The authors mention aligning the vertebrae based on virtually palpated anatomical landmarks and applying affine transformations, but the specific procedure used to reposition the vertebral bodies to achieve the target curvature lacks sufficient detail. Similarly, the method for anisotropic scaling of the vertebrae is not fully explained. Although affine transformations are said to be computed using specific landmarks, a more precise and step-by-step description would significantly improve the manuscript.

Authors: The relevant section in the methodology has been expanded to include these details (lines 268-319):

“Following the virtual palpation, a scoliotic model was generated with the OpenSim libraries (18,45) (via a custom script). The workflow for this is as follows (S2 Supporting Information - Generation of scoliotic spine model):

1. The reference systems of the model and the CT data (virtual palpation) were aligned by rotating the virtually palpated anatomical landmarks to correspond with the model reference system (Y axis in OpenSim corresponding to the inferior-superior direction of the CT data) and positioning the markers and the model such that the apex of the sacrum is coincident with the global OpenSim reference system.

2. The model was scaled, specifically, the vertebrae and pelvis were scaled using affine transformations using the OpenSim scaling tool. Three scaling factors were calculated for each vertebra – corresponding to the anterior-posterior, inferior-superior, and right-left directions. The scaling factors were calculated as the ratio of the Euclidean distance between the existing landmarks on the model and the virtually palpated anatomical landmarks. Specifically, the anterior-posterior scaling factor was calculated using the mean position of the landmarks on the anterior- and posterior-most points of the inferior and superior endplates; the inferior-superior scaling factor was calculated using the mean position of the landmarks on the inferior endplate and the mean position of the markers on the superior endplate; the right-left scaling factor was calculated using the mean position of the landmarks on the right- and left-most points of the inferior and superior endplates. Their masses were scaled based on the total mass of the subject (subject-specific mass of 75kg was assumed). The sacrum was not scaled as an appropriate automatic landmark (apex of sacrum) could not be reliably virtually palpated on the CT scan due to the field of view.

3. The sacrum of the model was aligned to the subject specific sacrum. The sacral slope was sacral slope defined in the CT data as the vector between virtually palpated anatomical landmarks on the posterior- and anterior-most points of the promontory of the sacrum.

4. The joint poses of each IVJ were redefined following the ISB recommendation (57). For each joint this was implemented by first defining IVJ axes for the scoliotic subject. The right-left axis of the joint as the vector between the anatomical landmarks on the pedicles of the vertebrae and the joint origin (midpoint between the anatomical landmarks on the centre of the endplates immediately inferior and superior to the joint). Following this the anterior-posterior axis of the joint was defined as the average of the lines passing anteriorly through the anatomical landmarks on the midline of the endplates immediately inferior and superior to the joint and on a plane defined by the joint origin and the right-left axis. The inferior-superior axis was then defined as perpendicular to the right-left axis and anterior-posterior axis pointing cephalad. This defined the scoliotic joints in the global reference system of the model. The model joint definitions (origin and orientations) were then updated to correspond to the scoliotic joints.

5. The vertebrae were realigned. Following the definition of each scoliotic IVJ, the vertebrae in the model were realigned based on the vertebral body alignment in the CT data. To define the alignment of the vertebrae a reference system for the body of each vertebra was defined in the global reference system. The inferior-superior axis of the vertebra was defined as the vector between the mean position of the landmarks on the inferior endplate and the mean position of the landmarks on the superior endplate; the anterior-posterior axis was defined as the mean of the vectors defined by the landmarks on the midline of the inferior and superior endplate; the right-left axis was defined as the cross product of the inferior-superior axis and the anterior-posterior axis. Using this definition the vertebrae were aligned in the corresponding parent joint reference frame.

6. Finally, the bushing forces are realigned with the joint to avoid introducing preloads due to joint stiffness.”

Reviewer #1: The definition of the intervertebral joint based on ISB recommendations is appropriate; however, the implementation itself would benefit from a brief explanation. Additionally, the manuscript mentions the addition of linear and spring dampers in the form of bushings, but several critical aspects are not clarified: were the connectors inserted after the morphological adjustments? Otherwise a pre-load would be introduced due to the roto-translation of adjacent vertebrae following the new curvature. Were the bushings replaced in the same relative location of traditional spherical joints?

Authors: The Authors have added a brief description of the IVJ definition according to the ISB recommendations (lines 239 - 245).

“In brief, the axis defining right-left translation direction is defined by the direction of the line between the landmarks of the pedicles of the vertebrae; the axis defining the inferior-superior translation direction is defined by the lines that connect the landmark on the centre of the inferior and superior endplate of the vertebrae; the axis defining the anterior-posterior shear is then perpendicular to the other two axes; and the origin of the joint is the intersect of the lines passing through the landmark on the centre of the inferior and superior endplate of the two adjacent vertebrae.”

The bushings are added before the morphological adjustments; the Authors would like to especially thank the Reviewer for noticing this detail in the methodology as they are absolutely correct. The code has been updated with the addition of a function that realigns the bushing forces with the joint reference frames after the morphological adjustments thus removing any preloads. This has been clarified in the code (lines 318- 319):

“6. Finally, the bushing forces are realigned with the joint to avoid introducing preloads due to joint stiffness.”

Regarding bushing forces, it is important to note that these do not replace the joints rather they are as forces at the location of the joints, the authors hope this is clarified with the modifications to lines 182 - 186:

“The original IVJs that allowed for the three rotational DoF were modified to also allow for the three translational DoF (for a total of 6 DoF at each IVJ). A lumped parameter spring-damper element (referred to in OpenSim as a bushing element) was then added at each IVJ, in the same relative location as the joints, to model its mechanical properties, as done by Christophy et al. (2012) (46).”

Reviewer #1: Concerning the virtual palpation performed by the operators, it is unclear whether they were clinicians or engineers. This distinction is important, as it relates directly to the reliability of the process. Furthermore, it is essential to specify what instructions were given to the operators for the virtual palpation. Although the variability analyses are detailed and statistically thorough, without knowledge of the protocol followed by the operators, it is difficult to assess whether the observed reliability is due to the method itself or to the operators’ expertise.

Authors: The Authors have clarified that it was engineers that performed the virtual palpations following a protocol which has been added as supporting information (S1 Supporting Information - Virtual Palpation Protocol). The authors would like to stress that while a segmentation was used to aid the virtual palpation, this was not a detailed segmentation but one created by only applying a threshold, no further processing was done (lines 325-332):

“To assess the inter-operator variability, the palpation was performed following a fixed protocol (S1 Supporting Information - Virtual Palpation Protocol) on the CT scan combined with a simple segmentation of the scan achieved with a single, quick thresholding operation with no further processing. The palpations were performed by five operators (all engineers, three of whom had extensive experience working with the spine in ex vivo, in vivo, and in silico studies), who repeated the procedure three times each (inter-operator dataset). Additionally, to evaluate the intra-operator variability, two of these operators (both with experience working with the spine) performed 10 repetitions (intra-operator dataset).”

Reviewer #1: Model accuracy is primarily assessed based on vertebral center position and curve fitting. While this is a valuable metric, but how anatomically implausible results (e.g., muscle paths intersecting bone) were identified and addressed? Was this done purely via visual inspection, or was a systematic metric or validation method used?

Authors: There are two components of these models that can be assessed for geometrical and morphological accuracy of the models, firstly the accuracy of the bony structures ie. the position, size, and orientation of the vertebrae. These as the reviewer noted were assessed based on the vertebral centre position and curve fitting. The second component would be the geometrical and morphological accuracy of the soft tissues, in OpenSim models this corresponds only to the muscle paths (the cross-sectional area of a muscle would correspond to the muscle material properties). The accuracy of the muscle paths was only assessed with a visual inspection during the initial evaluation of the method (line 320). It was only used as an initial evaluation to confirm that the wrapping spheres placed on the vertebrae resulted in the intended behavior – ensuring that muscle-in-bone penetration could not happen. Secondly a more robust method to assess the muscle path accuracy was not used as a ground truth of the muscle paths was not available as the muscles were not defined well enough in the available CT data and no suitable MRI data was available to segment the muscles and functional test data was not available to perform indirect validations based on parameters (such as joint moments and contact forces) from simulation predictions. These details have been added to the limitations of the study (lines 693-698):

“Furthermore, the accuracy of the muscle paths was only evaluated with a visual inspection in terms of anatomical plausibility (no muscle-in-bone penetration). Assessment of the patient-specific muscle path accuracy was not possible as magnetic resonance imaging (MRI) data was not available to establish the ground truth of the muscle paths. Indirect validation of the muscle paths based on the on parameters (such as joint moments and contact forces) from simulation predictions (77) was not possible as functional test data was not available.”

Reviewer #1: A noticeable mismatch is observed between vertebrae L3 to L5 in Fig. 6. The lumbar vertebrae appear to be more spaced than expected, especially imagining the posterior region where the facet joints are located. It would be important to clarify whether this discrepancy is due to the scaling procedure, landmark definition, or reference alignment. Was this a visual artifact? How the authors think this could reflect a recurring issue in the model output?

Authors: The Authors believe this mismatch in the lumbar levels is most likely due to severely deformed L2 and L3 vertebra which can be seen in the gold standard manual segmentation. This makes virtual palpation of anatomical landmarks very challenging. Even when done well, the process for defining the IVJ (based on anatomical landmarks) assumes healthy vertebrae, as the geometry of these vertebrae is so different to that of a healthy vertebra, it calls into question the suitability of the IVJ definition for pathological vertebrae. The discussion has been expanded to address this (lines 628-636, 647-649):

“There are multiple factors that may contribute to these systematic errors. The use of generic vertebra geometries rather than subject-specific geometries. The geometries can differ substantially, especially at levels where severe deformation is present, such as at L2 and L3 in the case of the subject used in the current study (Fig 2). This would reduce the correspondence of the positioning of the virtual palpitations between the model and the CT scan. This would alter the scaling and joint definition and would affect the position of the vertebral centres. One such example of this can be seen in the mismatch between vertebrae L2 to L5 (Fig 6) where the deformation of the L3 vertebra reduced the clarity of the L2 inferior endplate and the L4 superior endplate in the CT scan.”

“It should also be considered that the definition of the joint orientation was based on a non-pathological spine, meaning this definition may not be suitable for severe scoliosis.”

Reviewer #1: In the results, lumbar lordosis is reported as a negative angle. While this is likely due to an inverted curvature, angle convention generally refer to positive angles. Moreover, in the table, some terminology doesn’t appear straightforwardly clear: what is intended for “main curve type” and “main secondary curve type” ? Do they refer to anatomical regions (e.g., thoracic vs. lumbar)? Similarly, the term “pelvic mismatch” is used without adequate context regarding its clinical significance or its impact on the model.

Authors: The Authors have updated the reported angles to respect the convention of using positive angles. The terminology refers to a commonly used clinical classification system – Lenke Classification. Reference to this has been added (line 263).

Additionally, the section in the discussion which addresses the spinopelvic parameters has been expanded to address the clinical relevance of these parameters (lines 680-689):

“Spinopelvic parameters (such as pelvic incidence, pelvic tilt, thoracic kyphosis, and lumbar lordosis mismatch – the difference between the pelvic incidence and the lumbar lordosis (72)) and sagittal balance are important parameters in surgical planning and decision making and influence the outcome of surgical corrections (72–76). Poor management of these parameters has been associated with poor post-operative outcomes, such as post-operative lower back pain (72), proximal junctional kyphosis (73,74), distal junction kyphosis (74), disk degeneration (74), mechanical complications (instrumentation failure) (75), compensatory mechanisms to maintain balance (resulting in greater energy expenditure) (75), and generally lower health-related quality of life (76). Thus, evaluation of a wide range of spinopelvic parameters and spinal alignment is important.”

Reviewer #1: The authors point out that the reconstructed models align well with CT

---

## [Decision Letter · Decision Letter 1]

23 Oct 2025

Generation of severely scoliotic subject-specific musculoskeletal models

PONE-D-25-23772R1

Dear Dr. Samuele Luca Gould

We’re pleased to inform you that your manuscript has been judged scientifically suitable for publication and will be formally accepted for publication once it meets all outstanding technical requirements.

Kind regards,

Muhammad Mohsin Khan

Academic Editor

PLOS ONE

Reviewers' comments:

Reviewer's Responses to Questions

**Comments to the Author**

Reviewer #1: All comments have been addressed

Reviewer #2: All comments have been addressed

2. Is the manuscript technically sound, and do the data support the conclusions?

Reviewer #1: Yes

Reviewer #2: Yes

3. Has the statistical analysis been performed appropriately and rigorously?

Reviewer #1: Yes

Reviewer #2: Yes

4. Have the authors made all data underlying the findings in their manuscript fully available?

Reviewer #1: Yes

Reviewer #2: Yes

5. Is the manuscript presented in an intelligible fashion and written in standard English?

Reviewer #1: Yes

Reviewer #2: Yes

Reviewer #1: (No Response)

Reviewer #2: (No Response)

**Do you want your identity to be public for this peer review?** For information about this choice, including consent withdrawal, please see our Privacy Policy

Reviewer #1: **Yes: ** Simone Borrelli

Reviewer #2: No

---

## [Editor Report · Acceptance letter]

PONE-D-25-23772R1

PLOS ONE

Dear Dr. Gould,

I'm pleased to inform you that your manuscript has been deemed suitable for publication in PLOS ONE. Congratulations! Your manuscript is now being handed over to our production team.

Kind regards,

on behalf of

Dr. Muhammad Mohsin Khan

Academic Editor

PLOS ONE